# Dowry demand, perception of wife-beating, decision making power and associated partner violence among married adolescent girls: A cross-sectional analytical study in India

**Shobhit Srivastava**[1], **Pradeep Kumar**[2], **T. Muhammad**[3]*, **Manideep Govindu**[4], **Waad Ali**[5]

1 Department of Mathematical Demography & Statistics, International Institute for Population Sciences, Mumbai, Maharashtra, India, 2 Research and Data Analysis, Population Council India Office, New Delhi, India, 3 Department of Human Development and Family Studies, Pennsylvania State University, University Park, PA, United States of America, 4 Karnataka Health Promotion Trust, Bengaluru, India, 5 Department of Geography, Sultan Qaboos University, Muscat, Oman

* mkt5742@psu.edu, muhammad.iips@gmail.com

**Data Availability Statement:** The study utilizes a secondary source of data that is freely available in

## Abstract

### Background

Violence against women is considered a fundamental violation of their human rights. According to the world health organization (WHO), one-third of women worldwide experience some form of intimate partner violence (IPV). This study aimed to investigate the relationship between dowry demand, perception of wife-beating, decision-making on work and household purchases and physical, sexual, and emotional violence against married adolescent girls in India by using a large dataset.

### Methods

Data from the Understanding the lives of adolescents and young adults (UDAYA) project survey were used in this study. The final sample size included 4893 married adolescent girls. Descriptive statistics and bivariate analyses were performed to examine preliminary results. Multivariate logistic regression analysis was used to analyze the associations between dowry demand, perception of wife-beating, decision-making and types of IPV.

### Results

The prevalence of emotional, physical, and sexual violence among married adolescent girls were 28.6%, 22.9%, and 26.1%, respectively. Approximately 44% of married adolescent girls have experienced some form of violence (emotional, physical, or sexual). The likelihood of experiencing violence was 3.64 times more likely among adolescent girls who reported that dowry was demanded by their in-laws than their counterparts [aOR: 3.64; CI: 3.05–4.35]. Moreover, married girls who justified wife-beating were more likely to face any violence than their counterparts [aOR: 1.56; CI: 1.28–1.90]. Similarly, adolescent girls whose work decisions were made by others had higher odds of experiencing any violence

**Funding:** The authors received no specific funding
for this work.

**Competing interests:** The authors have declared
that no competing interests exist.

than those who decided their work themselves/jointly with others [aOR: 1.34; CI: 1.07–1.68]. The odds of any violence were higher among adolescent girls whose decisions on household purchases were made by others compared to those who decided to make purchases themselves or jointly with others [aOR: 1.37; CI: 1.09–1.71].

## Conclusions

The findings revealed significant associations between dowry-demand, justification of wife-beating, decision making power and IPV among married adolescent girls, and suggest policies that help reduce violence related to the predominantly practiced dowry system in the country, and programs aimed at educating adolescent married girls about their rights against violence and empowering them to retain equal decision-making power within their families and reduce their vulnerability to domestic violence.

## Background

Violence against women is considered a fundamental violation of women's human rights [1]. Intimate partner violence (IPV) refers to violence that occurs primarily from adolescence and early adulthood onward, most often in the context of marriage or cohabitation, and includes physical, sexual, and emotional abuse as well as controlling behaviours [2]. This definition emphasizes the multifaceted nature of IPV, extending beyond physical and sexual harm, to include emotional damage and control. According to the World Health Organization, one-third of women worldwide experience some form of IPV [3].

In South Asia, IPV is a complex challenge linked to cultural dynamics including strong patriarchal values, women's subordination in the family and limited opportunities for women in education and the workforce [4, 5]. Personal experiences and societal factors play a crucial role in heightening the risk of IPV. Research links IPV in the region, including India, to exposure to childhood trauma and abuse and witnessing inter-parental violence [6, 7] and young adults' perceptions of gender roles [8]. In a study on IPV in India using NFHS-4 data of Indian couples, the prevalence rates of emotional violence, physical violence, and sexual violence were 33.15%, 13.23%, and 6.60%, respectively [9]. Furthermore, studies have highlighted the role of societal factors, such as village tolerance of abuse and women's status, in contributing to the prevalence of IPV in rural areas of India [10]. Additionally, interventions, such as couple-based programs, have been explored for the primary prevention of IPV in India, emphasizing the need for effective strategies to address this issue [11, 12]. These programs involve both partners in counselling and education to address and modify harmful behaviors and foster healthier relationships.

Dowry as an aspect of marriage transactions is recognized as a key factor that underpins violence against women [13]. Although dowry is prohibited in India under the Dowry Prohibition Act 1961 and subsequent Sections 304 B and 498A of the Indian Penal Code, it remains a common practice [14]. A national-representative study reported that 22.5% of the in-laws demanded dowry for marriage [15]. Further, motives behind the predominant dowry system in South Asia, both in price and bequest form, are heterogeneous and depend primarily on bride or groom characteristics [16]. Dowry, on the other hand, has been implicated in many forms of gender-based violence such as son preference and sex-selective abortions [17–19], and several forms of sexual and emotional violence [20]. Additionally, a recent study in India

suggests that the price of gold at the time of marriage and the amount of gold a woman receives as a dowry were significant predictors of domestic violence [21]. Some earlier studies from India found that inadequate or delayed payment of dowry was an important reason for the perpetration of violence by a husband or in-laws [22–24]. There are also reports of domestic violence against women whose dowries were deemed insufficient by their husbands or their husbands' families [15].

Along with dowry demand, various factors exert influence on IPV, such as perception of wife-beating, decision-making on work and household purchases. Studies in Nepal and Pakistan have highlighted the attitudinal acceptance of wife-beating and its association with domestic violence [25, 26]. Furthermore, research in Bangladesh and Nigeria has explored the relationship between female participation in household decision making and the justification of wife-beating, shedding light on the complex dynamics influencing IPV [27, 28]. Additionally, an instrumental variables analysis in India has examined the attitudes about wife-beating and its incidence in domestic violence, providing insights into the interplay between perceptions and actual occurrences of IPV [29]. Furthermore, studies have shown that men's reporting on certain decision-making domains, such as large household purchases and expenditure on husband's earnings, predicted the likelihood of women experiencing IPV, highlighting the significance of decision-making dynamics in the context of IPV [30].

Donta et al. (2016) found that women's empowerment, which is measured by decision-making power, freedom of movement, and not justifying wife-beating, was less likely to be a risk factor for domestic violence [31]. Additionally, it is revealed that a decline in the male-female wage gap may reduce domestic violence by improving the intra-household bargaining power of women [32]. Nevertheless, it has been documented that women's lack of autonomy could be a major determinant for domestic violence victimization [33]. Moreover, patriarchal gender relations and women's subordinate status after marriage as new brides create a conducive context for violence in their husbands' home [34]. A cross-country analysis of attitudes towards wife-beating in Asia revealed that acceptance of wife-beating among women was highest in India [35]. Other studies in India show that a large amount of violence is perpetrated by the wider household, including female in-laws [36, 37]. Furthermore, some of the socioeconomic and demographic characteristics of women, such as urban residence, older age, lower education, and lower family income, were also associated with the occurrence of domestic violence [38].

India has an alarming trend of dowry deaths that sees more than 7000 women dying every year as a result of harassment over dowry [39]. The highest number of dowry deaths were recorded in the two northern states of Uttar Pradesh and Bihar [40]. However, little is known about the nature of the problem and married adolescent girls' experiences of dowry demands and reduced decision-making power during the early years of their marriage, resulting in some form of violence. Since there is scant research on this subject, and adolescence is a period of increased vulnerability to several types of violence due to lack of awareness and maturation [41], in this study, we aimed to examine the association of dowry demand, perception of wife-beating and decision-making on work and household purchases with physical, sexual, or emotional violence against married adolescents in India. We hypothesized that the demand for dowry, justification of wife-beating and lack of decision-making on work and household purchases are associated with increased violence among married adolescent girls.

## Methodology

### Overall approach

Understanding the lives of adolescents and young adults (UDAYA) project survey data was used in this study. The UDAYA project is a research initiative focused on understanding the

aspirations of youth and the developmental challenges faced by young people in India. A survey was conducted in two Indian states namely, Uttar Pradesh and Bihar in 2016 by the Population Council under the guidance of the Ministry of Health and Family Welfare, Government of India [42]. UDAYA collected detailed information on family, media, assets acquired during adolescence, community environment, and quality of transitions to young adulthood indicators. The study treated urban and rural areas of the state as independent sampling domains, and therefore, took sample areas independently for each of these two domains [42]. The details of the sampling design, fieldwork, data collection and survey weights are available in the UDAYA Report [42].

## Sampling strategy

This study was based on a subsample of the survey, which included 5,226 adolescents representing married girls between the ages of 15 and 19 years. The final sample size for this study was 4893 married adolescent girls who began cohabiting. Cases in which Gauna was not formed were removed from the sample (n = 333). Gauna is a northern Indian custom and ceremony associated with the consummation of marriage. This is associated with the custom of child marriages. The ceremony takes place several years after marriage. Before the ceremony, the bride stays at her natal home. Marriage is considered only as a ritual union, and conjugal life begins only after Gauna; that is, marriage is consummated only after the Gauna ceremony [43].

**Variable description.** Outcome variables in this study are the types of IPV, including emotional, physical and sexual violence. Exposure variables were selected based on an extensive literature review [2, 44–46], and include dowry demand, perception over wife-beating, decision making on going for work and on household purchases, paid work, marital duration and various socio-demographic factors such as age, education, age of spouse, place of residence, caste, religion, wealth quintile and state.

*Outcome variables*. Emotional violence was measured using the question, 'Did your husband ever do something to humiliate you in front of others or threatened you to hurt or harm someone close to you?' The responses were coded as no (0) or yes (1). Physical violence was recorded as 1 'yes' if the husband slapped, twisted or pulled hair, pushed/shook or throw something, kicked dragged beaten, burnt on purpose, attacked with a knife to the respondent in the last 12 months and 0 'no,' otherwise. Sexual violence was defined as 'yes' if the husband ever forced the respondent to have sex and 'no'; otherwise. Any violence was recoded as 1 'yes' if the married girl experienced any type of violence (emotional/physical/sexual); otherwise, 0 for "no".

*Exposure variables*. Dowry demand by in-laws was assessed by asking, "Has anybody in your husband's family ever said that the dowry/gift/cash you brought was too little or that you did not bring anything?" The responses were recoded as yes (1) and no (0). Perception over wife-beating was assessed through the question, "There are times when a wife deserves to be beaten by her husband." The response options were yes and no recoded as justified (1) and not justified (0). Decision about an adolescent girl joining workforce or not was measured through the question, "Who mainly takes the decision about whether you should work or stay at home? Would you say you alone decide, or you jointly decided with others or others alone decide?" We combined 'respondent only' and 'jointly with others' into 'herself or jointly with others,' (0) while retaining 'others only' (1) as a separate category. Decision-making on the household purchases was measured through the question "Who mainly takes the decision about making major household purchases? Would you say you alone decide, or you jointly decided with others or others alone decide?" and recoded as 'herself or jointly with others,' (0) and 'others only' (1). Paid work in the last 12 months was recoded as no (0) and yes (1). Marital duration (in years) was recoded as one or less than one year, 2–3 years, and four or more years.

The age of the adolescent girls was grouped into two categories: 15–17 years and 18–19 years. Educational level was recoded as no education, 1–7 years, 8–9 years, or $\geq$ 10 years. The place of residence was provided in the survey as rural and urban areas. Caste was grouped as Scheduled Caste/Scheduled Tribe (SC/ST), Other Backward Class (OBC), and others. Religion was recoded as Hindu or non-Hindu. Wealth quintile was created based on household asset data on ownership of selected durable goods, including means of transportation, as well as data on access to several amenities [42]. The variable was recoded into five equal quintiles: poorest, poor, middle, richer, and richest.

## Statistical analysis

Descriptive statistics and bivariate analyses were performed to examine preliminary results. To analyze the association between the binary outcome variable (violence experienced by married adolescent girls) and other explanatory variables, the binary logistic regression method [47] was used. STATA [48] command svyset was used to adjust the complex analysis, which includes the adjustment of clustering and the stratum effect. Data analysis was performed after assigning survey weights that were available in the UDAYA datasets. Multicollinearity was measured using the variance inflation factor (VIF) and it was found no evidence of multicollinearity among the variables used.

## Results

The sociodemographic characteristics of the married adolescent girls aged 15–19 years are presented in Table 1. Approximately one-fourth of the sample population reported that dowry was demanded by their in-laws, and nearly 27 percent of the participants justified wife-beating. More than half of the adolescent girls decided herself/jointly with others about their work. Moreover, 42 percent of adolescent girls decided herself/jointly with others on the matters of household purchases. Only 11 percent of girls had paid jobs. A higher proportion of girls belonged to the 18–19 years age group, and about one-fourth of the married adolescent girls had 10 or more years of schooling.

The percentage distribution of the types of violence among married adolescent girls aged 15–19 years is presented in Table 2. The prevalence of emotional, physical, and sexual violence was 28.6%, 22.9%, and 26.1%, respectively, among married adolescent girls. Moreover, approximately 44% of married adolescent girls experienced any type of violence (emotional/physical/sexual). The prevalence of violence was significantly higher among girls who reported that dowry was demanded by their in-laws. Similarly, married girls who justified wife-beating reported more violence. Moreover, there was a positive relationship between the types of violence and marital duration among adolescent girls. For instance, married girls with longer marital durations faced more violence than their counterparts with shorter marital durations. Similarly, violence had a significant negative association with girls' educational level and wealth index. Violence was more reported by girls who had no education and also those who belonged to the poorest families whereas it was lowest among girls who had 10 or more years of schooling and those who belonged to the richest families.

Tables 3 and 4 present the unadjusted and adjusted estimates from the logistic regression analysis of types of violence among married adolescent girls aged 15–19 years. The likelihood of any violence was 3.66 times higher among adolescent girls who reported that dowry was demanded by their in-laws than their counterparts [aOR: 3.66; CI: 3.06–4.37]. Moreover, married girls who justified wife-beating were more likely to face any violence than their counterparts [aOR: 1.56; CI: 1.28–1.90]. Similarly, adolescent girls whose work decisions were made by others had higher odds of experiencing any violence than those who decided their work

**Table 1. Characteristics of married adolescent girls aged 15–19 years.**

| Background characteristics | Percentage | N |
|---|---|---|
| **Dowry demanded by in-laws** | | |
| No | 74.1 | 3,678 |
| Yes | 25.9 | 1,215 |
| **Perception over wife-beating** | | |
| Not justified | 73.3 | 3,608 |
| Justified | 26.7 | 1,285 |
| **Decision-making on going to work** | | |
| Herself or jointly with others | 53.5 | 2,652 |
| Others Only | 46.5 | 2,241 |
| **Decision-making on household purchases** | | |
| Herself or jointly with others | 41.9 | 2,057 |
| Others Only | 58.1 | 2,836 |
| **Paid work (last 12 months)** | | |
| No | 88.9 | 4,361 |
| Yes | 11.1 | 532 |
| **Marital duration (in years)** | | |
| ≤1 | 41.2 | 1,928 |
| 2–3 | 38.2 | 1,922 |
| ≥4 | 20.6 | 1,043 |
| **Age groups (in years)** | | |
| 15–17 | 26.6 | 1,271 |
| 18–19 | 73.4 | 3,622 |
| **Education level (in years)** | | |
| No education | 27.3 | 1,365 |
| 1–7 years | 23.5 | 1,112 |
| 8–9 years | 25.0 | 1,214 |
| 10 & above | 24.2 | 1,202 |
| **Age of spouse** | | |
| ≤ 21 years | 31.8 | 1,512 |
| 22–24 years | 35.3 | 1,734 |
| 25+ years | 22.2 | 1,184 |
| Don't Know | 10.7 | 463 |
| **Place of residence** | | |
| Urban | 14.8 | 1,880 |
| Rural | 85.2 | 3,013 |
| **Caste** | | |
| SC/ST | 28.6 | 1,407 |
| OBC | 60.7 | 2,978 |
| Others | 10.6 | 508 |
| **Religion** | | |
| Hindu | 84.1 | 4,097 |
| Non-Hindu | 15.9 | 796 |
| **Wealth quintile** | | |
| Poorest | 13.6 | 676 |
| Poorer | 19.6 | 874 |
| Middle | 23.4 | 1,069 |
| Richer | 25.5 | 1,224 |

*(Continued)*

**Table 1.** (Continued)

| Background characteristics | Percentage | N |
|---|---|---|
| Richest | 17.9 | 1,050 |
| **State** | | |
| Uttar Pradesh | 35.4 | 1,711 |
| Bihar | 64.6 | 3,182 |
| Total | 100.0 | 4893 |

SC/ST: Scheduled Caste/Scheduled Tribe; OBC: Other Backward Class

themselves/jointly with others [aOR: 1.34; CI: 1.07–1.68]. The odds of any violence were higher among adolescent girls whose decisions on household purchases were made by others compared to those who decided to make purchases themselves or jointly with others [aOR: 1.37; CI: 1.09–1.71]. A detailed analysis with unadjusted and adjusted estimates is presented in S1 Table in S1 File and S2 Table in S2 File.

## Discussion

The current findings lend support to our hypothesis. This study found that dowry-demand by in-laws, justification of wife-beating and lack of decision-making on work and household purchases are significantly associated with some form of IPV among married adolescent girls in India.

The age-old custom of dowry and gifts for husbands and in-laws is strongly related to increased violence against women in India [15, 45, 49]. Consistently, our findings suggest that married adolescent girls who reported that dowry was demanded by their in-laws during or after marriage were more likely to experience IPV. The body of research on how dowry practices intersect with power imbalances, economic stressors, and societal norms to influence the prevalence and severity of IPV [50–52], may, in some way, explain our findings on dowry demand and IPV among adolescent married girls. For example, when dowry demands are unaffordable or unmet, the risk of IPV may increase, exacerbating gender inequality [53]. The bride's family may face financial challenges when meeting dowery obligations. In certain situations, these difficulties may translate into the bride becoming reliant on her husband's family for financial stability. Such a reliance can sometimes lead to a bride feeling compelled to accept IPV. Dowry practices may also exacerbate unequal power dynamics within marriages by fostering a sense of control over brides in the groom's family. This could lead to a reduction in the bride's autonomy, and further perpetuate gender inequality and IPV.

The results further revealed an association between married adolescent girls' justification of wife-beating and their experiences of partner violence. This finding aligns with previous research [18, 39, 40], which showed that a significant portion of the study sample that justified wife-beating also reported experiencing partner violence. The findings indicate that married women in India who still hold on gender norms and accept wife-beating as deep social learning have a higher likelihood of being victims of partner violence, which often perpetuates the traditional sex-stereotyped roles in society. It is worth noting that numerous studies, including those of Graham (1994) [54], Stark and Flitcraft (1988) [55], and Walker (1979) [56], have also revealed that survivors of IPV may be subjected to emotional abuse and manipulation by their abusers. For example, Walker identified the "Cycle of Violence," which comprises three phases: tension building, acute battering, and honeymoon/reconciliation. During the tension-building phase, conflict ensues between partners, whereas the acute battering phase involves the actual

**Table 2. Percentage distribution of type of violence among married adolescent girls aged 15–19 years.**

| Background characteristics | Emotional Violence | | Physical Violence | | Sexual Violence | | Any Violence | |
|---|---|---|---|---|---|---|---|---|
| | Percentage | p-value | Percentage | p-value | Percentage | p-value | Percentage | p-value |
| **Dowry demanded by in-laws** | | 0.000 | | 0.000 | | 0.000 | | 0.000 |
| No | 19.7 | | 16.3 | | 21.4 | | 35.4 | |
| Yes | 54.0 | | 42.1 | | 39.6 | | 68.4 | |
| **Perception over wife-beating** | | 0.001 | | 0.000 | | 0.000 | | 0.000 |
| Not justified | 26.5 | | 20.6 | | 22.4 | | 40.6 | |
| Justified | 34.3 | | 29.5 | | 36.2 | | 53.3 | |
| **Decision-making on going to work** | | 0.000 | | 0.612 | | 0.089 | | 0.000 |
| Herself or jointly with others | 32.6 | | 22.6 | | 27.6 | | 48.1 | |
| Others Only | 24.0 | | 23.4 | | 24.3 | | 39.2 | |
| **Decision-making on household purchases** | | 0.000 | | 0.135 | | 0.112 | | 0.000 |
| Herself or jointly with others | 34.7 | | 24.3 | | 28.1 | | 50.3 | |
| Others Only | 24.2 | | 22.0 | | 24.6 | | 39.4 | |
| **Paid work (last 12 months)** | | 0.392 | | 0.000 | | 0.001 | | 0.002 |
| No | 28.3 | | 22.0 | | 25.2 | | 43.0 | |
| Yes | 30.8 | | 30.5 | | 32.9 | | 51.4 | |
| **Marital duration (in years)** | | 0.000 | | 0.000 | | 0.410 | | 0.000 |
| ≤1 | 17.8 | | 14.3 | | 25.3 | | 35.3 | |
| 2–3 | 30.2 | | 25.7 | | 25.8 | | 45.1 | |
| ≥4 | 47.3 | | 35.1 | | 28.3 | | 59.2 | |
| **Age groups (in years)** | | 0.962 | | 0.008 | | 0.043 | | 0.987 |
| 15–17 | 28.5 | | 19.7 | | 28.9 | | 44.0 | |
| 18–19 | 28.6 | | 24.1 | | 25.1 | | 44.0 | |
| **Education level (in years)** | | 0.000 | | 0.000 | | 0.000 | | 0.000 |
| No education | 33.6 | | 30.4 | | 29.1 | | 48.4 | |
| 1–7 years | 33.3 | | 26.4 | | 29.3 | | 50.5 | |
| 8–9 years | 28.6 | | 21.8 | | 26.4 | | 44.0 | |
| 10 & above | 18.4 | | 12.4 | | 19.3 | | 32.6 | |
| **Age of Spouse** | | 0.000 | | 0.050 | | 0.000 | | 0.000 |
| ≤ 21 years | 26.6 | | 21.6 | | 27.5 | | 42.3 | |
| 22–24 years | 28.6 | | 21.3 | | 22.0 | | 42.0 | |
| 25+ years | 28.5 | | 25.1 | | 27.7 | | 45.3 | |
| Don't Know | 34.7 | | 28.2 | | 32.3 | | 52.7 | |
| **Place of residence** | | 0.026 | | 0.700 | | 0.201 | | 0.045 |
| Urban | 24.1 | | 22.3 | | 23.4 | | 39.3 | |
| Rural | 29.4 | | 23.1 | | 26.6 | | 44.8 | |
| **Caste** | | 0.000 | | 0.000 | | 0.000 | | 0.000 |
| SC/ST | 31.4 | | 28.1 | | 32.4 | | 48.3 | |
| OBC | 29.5 | | 22.0 | | 23.9 | | 44.2 | |
| Others | 15.7 | | 14.1 | | 21.6 | | 30.7 | |
| **Religion** | | 0.406 | | 0.328 | | 0.763 | | 0.936 |
| Hindu | 29.0 | | 23.3 | | 26.2 | | 44 | |
| Non-Hindu | 26.7 | | 21.2 | | 25.4 | | 43.8 | |
| **Wealth quintile** | | 0.015 | | 0.000 | | 0.032 | | 0.001 |
| Poorest | 32.0 | | 29.1 | | 30.4 | | 49.1 | |
| Poorer | 31.8 | | 25.9 | | 28.4 | | 46.1 | |
| Middle | 30.3 | | 25.5 | | 27.2 | | 46.3 | |

*(Continued)*

**Table 2.** (Continued)

| Background characteristics | Emotional Violence | | Physical Violence | | Sexual Violence | | Any Violence | |
|---|---|---|---|---|---|---|---|---|
| | Percentage | p-value | Percentage | p-value | Percentage | p-value | Percentage | p-value |
| Richer | 27.8 | | 19.6 | | 24.1 | | 43.7 | |
| Richest | 21.4 | | 16.5 | | 21.7 | | 35.0 | |
| **State** | | 0.000 | | 0.024 | | 0.033 | | 0.000 |
| Uttar Pradesh | 18.9 | | 20.2 | | 22.6 | | 36.9 | |
| Bihar | 33.9 | | 24.5 | | 28.0 | | 47.9 | |
| **Total** | 28.6 | | 22.9 | | 26.1 | | 44.0 | |

SC/ST: Scheduled Caste/Scheduled Tribe; OBC: Other Backward Class

occurrence of violence. Finally, during the honeymoon/reconciliation phase, the perpetrator reconciles, further misleading IPV survivors into believing that abuse will not recur. IPV perpetrators employ suite tactics such as gaslighting, victim-blaming, and minimizing abuse to manipulate IPV survivors to accept inflicted abuse [56]. Other studies have suggested that IPV survivors may use coping mechanisms, such as cognitive dissonance, to rationalize the abuse they experience [55]. Similarly, Graham's work describes the "Stockholm Syndrome" as a coping mechanism whereby IPV survivors develop an attachment to their abusers, defend their actions, and ultimately believe that they deserve inflicted abuse [54]. These theories support our finding and highlight an important area of future research to effectively address the justification of wife-beating among married adolescent girls in India and reduce the incidence of IPV among them.

Previous studies have also shown that power dynamics in the family and women's empowerment have a greater impact on women's status, such that the chances of being mistreated are lower among women involved in the family's decision-making processes [46, 57]. Consistently, the present study showed that married adolescent girls who had no role in decision making on matters of her going for work and household purchases were significantly more likely to

**Table 3. Unadjusted estimates (Crude ORs) from logistic regression analysis of types of violence among married adolescent girls aged 15–19 years.**

| Background characteristics | Emotional Violence | Physical Violence | Sexual Violence | Any Violence |
|---|---|---|---|---|
| | Crude OR (CI 95%) | Crude OR (CI 95%) | Crude OR (CI 95%) | Crude OR (CI 95%) |
| **Dowry demanded by in-laws** | | | | |
| No | | | | |
| Yes | 4.32[3.42–5.45]*** | 3.62[2.88–4.54]*** | 2.32[1.9–2.85]*** | 3.64[3.05–4.35]*** |
| **Perception over wife-beating** | | | | |
| Not justified | | | | |
| Justified | 1.3[1.04–1.62]** | 1.36[1.07–1.73]** | 1.82[1.48–2.24]*** | 1.55[1.27–1.88]*** |
| **Decision-making on going to work** | | | | |
| Herself or jointly with others | | | | |
| Others Only | 0.69[0.54–0.88]*** | 1.06[0.84–1.33] | 0.84[0.67–1.05] | 0.74[0.59–0.92]** |
| **Decision-making on household purchases** | | | | |
| Herself or jointly with others | | | | |
| Others Only | 0.69[0.54–0.89]*** | 0.91[0.72–1.16] | 0.88[0.69–1.12] | 0.72[0.58–0.9]*** |

***p<0.001

**p<0.05

*p<0.10; aOR: Adjusted Odds Ratio; CI: Confidence Interval

**Table 4. Adjusted estimates (adjusted ORs) from logistic regression analysis of types of violence among married adolescent girls aged 15–19 years.**

| Background characteristics | Emotional Violence aOR (CI 95%) | Physical Violence aOR (CI 95%) | Sexual Violence aOR (CI 95%) | Any Violence aOR (CI 95%) |
|---|---|---|---|---|
| **Dowry demanded by in-laws** | | | | |
| No | | | | |
| Yes | 4.35 [3.44–5.48]*** | 3.63 [2.89–4.57]*** | 2.33 [1.89–2.86]*** | 3.66 [3.06–4.37]*** |
| **Perception over wife-beating** | | | | |
| Not justified | | | | |
| Justified | 1.30 [1.04–1.62]** | 1.38 [1.08–1.76]** | 1.85 [1.5–2.29]*** | 1.56 [1.28–1.9]*** |
| **Decision-making on going to work** | | | | |
| Herself or jointly with others | | | | |
| Others Only | 1.44 [1.12–1.83]*** | 0.94 [0.75–1.18] | 1.18 [0.95–1.48] | 1.34 [1.07–1.68]** |
| **Decision-making on household purchases** | | | | |
| Herself or jointly with others | | | | |
| Others Only | 1.44 [1.12–1.86]** | 1.09 [0.85–1.39] | 1.12 [0.87–1.44] | 1.37 [1.09–1.71]** |

***p<0.001

**p<0.05

*p<0.10; aOR: Adjusted Odds Ratio; CI: Confidence Interval; The aOR are adjusted for Paid work (last 12 months), Marital duration (in years), Age groups (in years), Education level (in years), Place of residence, Caste, Religion, Wealth quintile, State

experience some type of violence. Furthermore, it is shown that employed women may have more domestic power and are more likely to be exposed to media, and their financial resources allow them to question traditional subordinate roles and practices [58]. A study in Uttar Pradesh found that women's participation in paid work and house ownership was significantly associated with a reduction in experiencing marital violence [59]. However, as far as paid work among married adolescent girls in the present study is concerned, the protective effect of women's employment on the violence that is shown in several past studies [60, 61], was not found in the present study. The analysis showed that adolescent girls who are engaged in paid work are significantly more likely to experience physical or sexual violence than their counterparts. The finding is similar to a study based on the NFHS data which noted that because of being forthright against male dominance and better reporting of incidences of domestic violence by which they pose a threat to their partners on traditional gender roles and hence, Indian working women are vulnerable to partner violence [62]. Again, women who are engaged in small businesses and farming, have a higher economic status than their husbands, and are seen as having sufficient power to change traditional gender roles are at greater risk of experiencing partner violence [63–65]. This finding may be explained by the fact that economically empowered women who have more gender-conservative partners may have an increased risk of experiencing violence, especially when they become less willing to conform to household patriarchal norms [66].

Finally, evidence suggests that status and power discrepancies in owning resources and young women's lack of education may be associated with increased perpetration of domestic violence, especially among men who hold traditional ideas about gender and believe that men should be the primary breadwinners for a family [67]. A study found that uneducated women were more likely to marry early, and a lack of education exacerbated their disadvantages in society [68]. A wealth of literature has repeatedly found educational attainment and higher social status to be major factors in reducing the likelihood of experiencing domestic violence in developed as well as developing countries [69–72]. The findings from this study establish

that higher education and upper social class in the Indian caste hierarchy of married adolescent girls were protective factors against partner violence.

The present study has certain limitations to be acknowledged. Owing to the cross-sectional nature of this study, no causal relationships between dowry demand, perception of wife-beating, decision making power and IPV could be established. Additionally, while this study did not explicitly examine the stratification of personal, social, and cultural factors influencing IPV, existing literature indicates that cultural factors have a more pronounced impact on IPV compared to personal or social factors. For instance, a study conducted among Asian Indian population groups in the US revealed that traditional cultural norms and beliefs reinforcing gender-based roles contribute to attitudes that perpetuate IPV, underscoring the significance of challenging these deeply ingrained cultural values to mitigate IPV [73]. These factors should be considered in future research.

Another major limitation of the study is that all data were self-reported by married adolescent girls and therefore subject to recall and social desirability biases, particularly those related to the private realm, such as dowry demand by husbands or in-laws and experiencing domestic violence. These biases could potentially result in an underestimation of violence experienced by married adolescent girls. It is possible that the underreporting of violence among married adolescent females may be attributed to their hesitance to disclose personal and/or stigmatized experiences in their relationship with their partners. Therefore, the reported prevalence of violence may not accurately reflect the actual incidence within this demographic group.

Similarly, the findings may also be subject to selection bias, because the data did not capture the worst cases of domestic violence in the sample. Future research could benefit from employing mechanistic qualitative studies, employing a suite of methods, such as in-depth interviews, focus groups, and participant observation, to explore the link between early marriage among adolescent girls and IPV. By doing so, a more comprehensive understanding of the lived experiences, perceptions, and dynamics of adolescent girls who have experienced both early marriage and IPV can be achieved. Moreover, considering the focus of the current study on partner violence, it is crucial for future studies to explore other forms of violence that may be inflicted by members of the same family, such as sexual violence or dowry-related violence. Despite these limitations, this study sheds light on the potential determinants of IPV in India and contributes to the existing literature on domestic violence because of its large population-based data.

## Conclusions

This study found significant relationships between dowry demand, justification of wife-beating, reduced decision-making power and physical, sexual, or emotional violence against married adolescent girls, thereby offering a more comprehensive understanding of how strategies could be developed to better address an important public health issue of IPV among married adolescent girls in India.

Taken together, the findings of this study can be placed within the context of how dowry practices may exacerbate gender inequality through the manifestation of dowry demands, acceptance of IPV, and the absence of decision-making power among women. The practice of demanding dowry may exacerbate women's devaluation and perpetuate unequal power dynamics in marital relationships. To address gender inequality, as it manifests in dowry demands, acceptance of IPV, and lack of female decision-making power, a holistic approach that encompasses challenging conventional gender norms and promoting women's empowerment is needed. Concerning the latter, interventions such as educational initiatives that equip married adolescents with essential information to recognize and resist dowry-related violence

and advocating for legal reform to discourage such violence could collectively contribute to mitigating the occurrence of IPV among married adolescents and empowering them. The findings also have significant policy implications as it can aid in the development of prudent gender-sensitive policies that target the root causes of gender inequality, as it relates to the structural inequalities of dowry practices.

## Supporting information

**S1 File.** S1 Table. Unadjusted estimates (Crude ORs) from logistic regression analysis of types of violence among married adolescent girls aged 15–19 years.
(DOCX)

**S2 File.** S2 Table. Adjusted estimates (adjusted ORs) from logistic regression analysis of types of violence among married adolescent girls aged 15–19 years.
(DOCX)

## Author Contributions

**Conceptualization:** Shobhit Srivastava, Pradeep Kumar, T. Muhammad, Waad Ali.

**Data curation:** Shobhit Srivastava, T. Muhammad, Manideep Govindu.

**Formal analysis:** Shobhit Srivastava, Pradeep Kumar, Manideep Govindu.

**Investigation:** Shobhit Srivastava, T. Muhammad, Manideep Govindu.

**Methodology:** Shobhit Srivastava, Pradeep Kumar, T. Muhammad.

**Supervision:** Pradeep Kumar, T. Muhammad, Waad Ali.

**Validation:** Shobhit Srivastava, Pradeep Kumar, T. Muhammad, Waad Ali.

**Writing – original draft:** Shobhit Srivastava, Pradeep Kumar, T. Muhammad, Waad Ali.

**Writing – review & editing:** Shobhit Srivastava, Pradeep Kumar, T. Muhammad, Manideep Govindu, Waad Ali.

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
