## [Decision Letter · Decision Letter 0]

9 Nov 2021

PONE-D-21-13207

Dowry demand and associated partner violence among married adolescent girls: A cross-sectional analytical study in India

PLOS ONE

Dear Dr. T.,

Thank you for submitting your manuscript to PLOS ONE. After careful consideration, we feel that it has merit but does not fully meet PLOS ONE’s publication criteria as it currently stands. Therefore, we invite you to submit a revised version of the manuscript that addresses the points raised during the review process.

We look forward to receiving your revised manuscript.

Kind regards,

Susan A. Bartels, MD, MPH, FRCPC

Academic Editor

PLOS ONE

Journal Requirements:

Reviewers' comments:

Reviewer's Responses to Questions

**Comments to the Author**

1. Is the manuscript technically sound, and do the data support the conclusions?

Reviewer #1: Yes

Reviewer #2: No

2. Has the statistical analysis been performed appropriately and rigorously? 

Reviewer #1: Yes

Reviewer #2: No

3. Have the authors made all data underlying the findings in their manuscript fully available?

Reviewer #1: Yes

Reviewer #2: Yes

4. Is the manuscript presented in an intelligible fashion and written in standard English?

Reviewer #1: Yes

Reviewer #2: Yes

5. Review Comments to the Author

Reviewer #1: The research and resulting manuscript is very well done. Please complete minor editorial changes (e.g. data should always be plural, "data are" not "is"). Also, is there a possibility for comparison of data from other Indian states? That would be an interesting next step.

Reviewer #2: Referee Report on “Dowry demand and associated partner violence among married adolescent girls: A cross-sectional analytical study in India”

This paper explores the relationship between dowry demand and domestic violence in two states in India. The methodology is descriptive, and the authors find that the prevalence of domestic violence was higher amongst girls who reported that dowry was demanded by their husbands. While, an important topic with nuanced relationships this paper does not make a sufficient enough contribution beyond the existing literature. Here are the main reasons why:

1. The paper is based on data from the UDAYA project. The sole advantage of this data over more comprehensive and nationally representative surveys such as the DHS which records domestic violence, is that an additional question regarding dowry demand was asked. Why did the authors not use the India Human Development Survey (IHDS) panel dataset which has even more detailed dowry and domestic violence information and more coverage with regards to age and geography?

2. The authors are particularly interested in examining this amongst adolescent girls, but the paper lacks any motivation as to why this would be interesting and relevant to examine.

3. In the subsection labelled “Data,” the authors elaborate on how they select their analytical sample. However, the numbers do not add up. According to the information provided, the analytical sample of married girls from both states should be 5206 (1798 + 3408) and not 5226. Similarly, the number for the effective sample size do not add up either and the number of observations used for the estimations are not provided in the tables.

4. The equation in the subsection titled “Statistical Analysis” is not explained at all and there is no discussion of the perils of using a logistic regression to estimate this relationship or that of sample selection (the worst cases of domestic violence are likely not captured by the sample) which are surely important considerations when examining this relationship. Furthermore, there is no discussion of the endogeneity of dowry demanded which is certain to make this estimate biased.

5. Despite these shortcomings, there are some very crucial claims made by the paper which are unsubstantiated. For instance, the sentence “In the process of empowering women, the study indicates that parents appear to be replacing property inheritance to their daughters by giving them alternative transfers in the form of higher dowries.” The authors do not have any information on inheritance and do not present any results for such a replacement. ‘

6. Another such claim in the Conclusion section is “The results presented in this study suggest that policies that ensure equal inheritance and property rights for women and programs that help women retain equal power and say in their families may be necessary to reduce their vulnerability to domestic violence.” The authors have presented no evidence to support this claim.

6. PLOS authors have the option to publish the peer review history of their article (what does this mean?). If published, this will include your full peer review and any attached files.

Reviewer #1: **Yes: **Priya Banerjee

Reviewer #2: No

---

## [Author Response · Author response to Decision Letter 0]

23 Nov 2021

Response to comments

Reviewer #1: 

The research and resulting manuscript is very well done. Please complete minor editorial changes (e.g. data should always be plural, "data are" not "is"). 

Response: Thanks for the suggestion. Changes have been made.

Also, is there a possibility for comparison of data from other Indian states? That would be an interesting next step.

Response: The survey was done in only these two states: Bihar and Uttar Pradesh therefore author unable to compare data with other Indian states. 

Reviewer #2: 

Referee Report on “Dowry demand and associated partner violence among married adolescent girls: A cross-sectional analytical study in India”

This paper explores the relationship between dowry demand and domestic violence in two states in India. The methodology is descriptive, and the authors find that the prevalence of domestic violence was higher amongst girls who reported that dowry was demanded by their husbands. While, an important topic with nuanced relationships this paper does not make a sufficient enough contribution beyond the existing literature. Here are the main reasons why:

1. The paper is based on data from the UDAYA project. The sole advantage of this data over more comprehensive and nationally representative surveys such as the DHS which records domestic violence, is that an additional question regarding dowry demand was asked. Why did the authors not use the India Human Development Survey (IHDS) panel dataset which has even more detailed dowry and domestic violence information and more coverage with regards to age and geography?

Response: Dear reviewer, I agree with your comment that India Human Development Survey (IHDS) panel dataset which has even more detailed dowry and domestic violence information and more coverage with regards to age and geography. However, there are certain limitations i.e., the data is very old (2004-05 and 2011-12), therefore the findings cannot be generalised in today’s context. The data was also not available at district level. The authors used UDAYA data because firstly, the authors aim to determine the association between IPV and dowry among adolescent girls. Secondly; in spite of the fact that the data was available only for two states, the sample is representative and also the estimates are of recent times which can be generalized for UP and Bihar. 

Still authors on the suggestion of reviewer checked IHDS data, it was found that the sample was too low for 15-19 age group females and additionally, the prevalence of dowry was 100%. Therefore, analysing the relationship was not possible. 

2. The authors are particularly interested in examining this amongst adolescent girls, but the paper lacks any motivation as to why this would be interesting and relevant to examine.

Response: The ‘adolescent girls’ is a vulnerable group with lack of awareness, maturity and need more intervention. Adolescence therefore, is also a period of increased vulnerability to several types of violence. These are mentioned in the revised version.

3. In the subsection labelled “Data,” the authors elaborate on how they select their analytical sample. However, the numbers do not add up. According to the information provided, the analytical sample of married girls from both states should be 5206 (1798 + 3408) and not 5226. Similarly, the number for the effective sample size do not add up either and the number of observations used for the estimations are not provided in the tables.

Response: In page number 8 there was an error it was 313 cases were removed.

We have to provide the N in other tables.

Bihar sample 

Married women was 3408 samples and if we remove 226 then 3182 which was matching our tables similarity UP states also. 

226 + 87�313. The cases for which Gauna was not formed were removed from the sample (313 cases) 

UP sample 

State 

Uttar Pradesh 35.4 1,711

Bihar 64.6 3,182

Total 100.0 4893

4. The equation in the subsection titled “Statistical Analysis” is not explained at all and there is no discussion of the perils of using a logistic regression to estimate this relationship or that of sample selection (the worst cases of domestic violence are likely not captured by the sample) which are surely important considerations when examining this relationship. Furthermore, there is no discussion of the endogeneity of dowry demanded which is certain to make this estimate biased.

Response: Dear reviewer, the data do not capture the worst cases of domestic violence. This is now mentioned in the limitation. The equation is explained in the revised version. The issue of endogeneity was looked upon before the analysis; however, it was not mentioned in the manuscript. The authors now have mentioned the same in the manuscript. 

5. Despite these shortcomings, there are some very crucial claims made by the paper which are unsubstantiated. For instance, the sentence “In the process of empowering women, the study indicates that parents appear to be replacing property inheritance to their daughters by giving them alternative transfers in the form of higher dowries.” The authors do not have any information on inheritance and do not present any results for such a replacement. ‘

Response: The sentence is modified. The statement is related to a previous study which partly explains the less violence among empowered adolescent girls in our study which was mentioned in the next line.

6. Another such claim in the Conclusion section is “The results presented in this study suggest that policies that ensure equal inheritance and property rights for women and programs that help women retain equal power and say in their families may be necessary to reduce their vulnerability to domestic violence.” The authors have presented no evidence to support this claim.

Response: Thank you for pointing this. The statements have been revised accordingly.

---

## [Decision Letter · Decision Letter 1]

9 Jun 2022

PONE-D-21-13207R1Dowry demand and associated partner violence among married adolescent girls: A cross-sectional analytical study in IndiaPLOS ONE

Dear Dr. T.,

Thank you for submitting your manuscript to PLOS ONE. After careful consideration, we feel that it has merit but does not fully meet PLOS ONE’s publication criteria as it currently stands. Therefore, we invite you to submit a revised version of the manuscript that addresses the points raised during the review process.

We look forward to receiving your revised manuscript.

Kind regards,

Susan A. Bartels, MD, MPH, FRCPC

Academic Editor

PLOS ONE

Reviewers' comments:

Reviewer's Responses to Questions

**Comments to the Author**

1. If the authors have adequately addressed your comments raised in a previous round of review and you feel that this manuscript is now acceptable for publication, you may indicate that here to bypass the “Comments to the Author” section, enter your conflict of interest statement in the “Confidential to Editor” section, and submit your "Accept" recommendation.

Reviewer #3: (No Response)

Reviewer #4: (No Response)

2. Is the manuscript technically sound, and do the data support the conclusions?

Reviewer #3: Yes

Reviewer #4: Yes

3. Has the statistical analysis been performed appropriately and rigorously? 

Reviewer #3: Yes

Reviewer #4: I Don't Know

4. Have the authors made all data underlying the findings in their manuscript fully available?

Reviewer #3: Yes

Reviewer #4: No

5. Is the manuscript presented in an intelligible fashion and written in standard English?

Reviewer #3: Yes

Reviewer #4: Yes

6. Review Comments to the Author

Reviewer #3: This paper examines the relationship between dowry demands and IPV among adolescent girls. I have a few suggestions for improvement:

1) Throughout the paper, I think the language of risk factors need to be changed to correlates since this is a cross sectional study. Also calling a paid job and marital duration as risk factors could present them as factors that need to be corrected for when findings on employment among women related to IPV in India are mixed. The empowerment of women can also sometimes protect women. Being in marriage for a long time can also be a reason why women are facing violence. Authors elaborate on this in the discussion but I think correcting the language early on would be better. This could be changed in the abstract and other areas of the paper.

2) In introduction, you could include how you define intimate partner violence- method section discusses that but will be good for the reader to know early on.

3) Methods: the sentence on page 19 needs to be corrected--"For instance, married girls faced more violence to increase the length of the marriage."

4) Discussion- this section could be strengthened with discussing how early marriage in adolescence can enhance risk and the practice and policy implications of early marriage

Reviewer #4: This is an interesting paper which explores the relationship between demands for dowry and domestic violence focusing on young women which is an important topic. However, the paper raises a number of questions and comments – which if addressed might make for an interesting contribution to the literature in this field.

1. The hypothesis that dowry demands are associated with increased violence against women is neither new nor particularly helpful, unless the authors explain what their study contributes to existing knowledge and why it is important.

2. The conceptual framework is a bit simplistic – it simply connects a number of facts and says they lead to increased violence. It is not clear if all the elements in the first box lead to all the elements in the second box, and regardless they lead to violence? I would have thought that some of the factors in the second box would be protective factors, i.e. paid work and decision making autonomy – or are the authors proposing that despite the existence of these facts – that all of them are necessary for violence against women? There are a few interesting ideas here, but I would encourage the authors to create a more nuanced conceptual framework – where some factors may promote and others protect against perpetration of violence. Also some factors may produce violence acting alone (e.g. demands for dowry) and others in combination (social class plus attitudes towards women) to suggest increased likelihood of violence. The internal relationship between these identified factors needs to be reflected in the figure presented.

3. Strongly suggest that the figures and tables are labelled. It would also be good to have a table giving an overview of the relevant demographic information about the sample of participants in one glance.

4. Outcome variable for all types of violence focused only on violence by the husband, however, often violence (dowry related or otherwise) including sexual violence, is inflicted other members of the family – how do the authors account for that? At the very least some acknowledgement of the fact is required.

5. Similarly, they continue to say, “Similarly, violence had a significant positive association with the educational level of girls and wealth index.” – however it is a negative relationship between violence and educational level – i.e. the higher the education, the lower the violence and the same with wealth. The way ‘positive association’ reads now it implies, the greater the wealth the more the violence inflicted – which is the opposite of what the authors go on to explain is happening.

6. The authors report, “Interestingly, the likelihood of any violence was 41 percent and 2.06 times more likely among adolescent girls whose marital duration was 2-3 years [OR: 1.41; CI: 1.19-1.66] and 4 years or more [OR: 2.06; CI: 1.60- 2.64] respectively, compared to those whose marital duration was less than or equal to one year.” Why is this interesting? Is it not obvious – the more exposure, the more likelihood of experiencing violence? What would be interesting would be to explore if the trajectory of violence is upward with time and yet the girls choose to remain in the marriage? Is there a tipping point at which the girl might take some action or complain? Probably the data does not allow the authors to answer these questions, but they are nevertheless more interesting.

7. In the discussion section, the authors report: “In the process of empowering women, the previous study indicates that parents appear to be replacing property inheritance to their daughters by giving them alternative transfers in the form of higher dowries (42). This might have resulted in husbands and family in-laws of empowered women not perpetrating any violence.” – that is a big generalisation – often dowries are taken over by the in-laws and the woman has little or no control over her wealth. What is the evidence that higher dowries succeed in empowering women – especially if the amount bestowed is lower than that demanded by the in-laws?

8. Furthermore, the authors report, “The finding is similar to a study based on the National Family Health Survey which noted that because of being forthright against male dominance and better reporting of incidences of domestic violence, Indian working women have a higher likelihood of being victims of violence (47).”

It is not clear what the authors are claiming from their finding that those in paid employment are more likely to be victims of domestic violence – is it because they are more willing to acknowledge and report violence or because they pose a threat to their partners on traditional gender roles and hence are vulnerable to partner violence?

Further, does this mean that having paid employment makes some women more vulnerable and not having enough economic empowerment makes other women more vulnerable? If in both situations some women are vulnerable – then there must be other factors that will determine who is more likely to be vulnerable with economic empowerment and who isn’t. On the one hand the authors seem to suggest in the Discussion that when women are more likely to report violence (either because they are economically empowered or have been married for a longer period) they are more likely to suffer violence, but do not mention why this might be the case. Are the findings suggesting that women who are empowered to speak out against violence are more likely to suffer future violence as a result – if so – this sends out a very negative message as it does not support more empowerment for women – perhaps the authors might need to carefully caveat this finding.

Further they go on to state that higher education status and social class act as protective factors and such women are less likely to be victims of violence. I am not sure the authors have reconciled these contrary findings effectively.

9. A few grammatical issues need addressing. For example,

a. There are no page numbers for giving exact location, but in the results section the authors report “For instance, married girls faced more violence to increase the length of the marriage.” – which does not sound correct – Suggest rephrase.

b. “A dearth of literature repeatedly found educational attainment and higher social status as major factors to reduce the likelihood of experiencing domestic violence in developed as well as developing countries (56–59)”. How can a dearth of literature (which means lack of literature) repeatedly find ….something?

c. The authors state, “This study is based on sample of married girls from both the states which is 5,226 which is representative of 15-19 married girls at state level.” – perhaps they mean sample representing married girls between the age group of 15 to 19 years.

10. One big gap in the paper is the relationship between violence and whether the woman has been able to produce children, preferably male offspring which might be related to or completely separate to demands for dowry. Were the women asked whether they felt their victimisation by their partner was due to unfulfilled demands for dowry and/or other factors?

11. Did the dataset not have any information about dowry related domestic violence not perpetrated by the husband/partner? How do the authors propose to deal with this lacuna?

I do believe that the paper can make an important contribution if the authors work on developing a more nuanced conceptual framework and hypothesise which factors provide effective protection or which combination of factors make women more vulnerable to victimisation. A defined theoretical framework to make better sense of the data would make a significant contribution to the current literature on the topic. This would also help the authors structure their answer the ‘so what’ question better in terms of concrete policy implications which in the current version are a bit imprecise.

7. PLOS authors have the option to publish the peer review history of their article (what does this mean?). If published, this will include your full peer review and any attached files.

Reviewer #3: No

Reviewer #4: **Yes: **Jyoti Belur

---

## [Author Response · Author response to Decision Letter 1]

6 Jul 2022

Response to comments

Reviewer #3: 

This paper examines the relationship between dowry demands and IPV among adolescent girls. I have a few suggestions for improvement:

1) Throughout the paper, I think the language of risk factors need to be changed to correlates since this is a cross sectional study. Also calling a paid job and marital duration as risk factors could present them as factors that need to be corrected for when findings on employment among women related to IPV in India are mixed. The empowerment of women can also sometimes protect women. Being in marriage for a long time can also be a reason why women are facing violence. Authors elaborate on this in the discussion but I think correcting the language early on would be better. This could be changed in the abstract and other areas of the paper.

Response: The amendments are made as per suggestion.

2) In introduction, you could include how you define intimate partner violence- method section discusses that but will be good for the reader to know early on.

Response: The definition of IPV is provided in the introduction section of the revised version.

3) Methods: the sentence on page 19 needs to be corrected--"For instance, married girls faced more violence to increase the length of the marriage."

Response: The statement is modified for better understanding.

4) Discussion- this section could be strengthened with discussing how early marriage in adolescence can enhance risk and the practice and policy implications of early marriage

Response: Dear reviewer, thank you for the comment. Since the current study focused on married girls aged 15 to 19 years and not above that, the authors feel that discussing on early marriage will be superficial. Even, the age groups of 15-17 and 18-19 years showed no statistically significant association with most of the outcome variables. However, this is acknowledged in detail in limitations of the study with the suggestion of future studies on this aspect.

Reviewer #4: 

This is an interesting paper which explores the relationship between demands for dowry and domestic violence focusing on young women which is an important topic. However, the paper raises a number of questions and comments – which if addressed might make for an interesting contribution to the literature in this field.

1. The hypothesis that dowry demands are associated with increased violence against women is neither new nor particularly helpful, unless the authors explain what their study contributes to existing knowledge and why it is important.

Response: The last paragraph of the introduction section is revised accordingly and added the potential contribution of the study.

2. The conceptual framework is a bit simplistic – it simply connects a number of facts and says they lead to increased violence. It is not clear if all the elements in the first box lead to all the elements in the second box, and regardless they lead to violence? I would have thought that some of the factors in the second box would be protective factors, i.e. paid work and decision making autonomy – or are the authors proposing that despite the existence of these facts – that all of them are necessary for violence against women? There are a few interesting ideas here, but I would encourage the authors to create a more nuanced conceptual framework – where some factors may promote and others protect against perpetration of violence. Also some factors may produce violence acting alone (e.g. demands for dowry) and others in combination (social class plus attitudes towards women) to suggest increased likelihood of violence. The internal relationship between these identified factors needs to be reflected in the figure presented.

Response: Thank you for the detailed comment. The revised conceptual framework has been provided which illustrates the risk factors and protective factors separately and the direct and combined effects on IPV.

3. Strongly suggest that the figures and tables are labelled. It would also be good to have a table giving an overview of the relevant demographic information about the sample of participants in one glance.

Response: The tables are labeled already as table 1, 2 and 3. The demographic information of the sample population is included in Table 1.

4. Outcome variable for all types of violence focused only on violence by the husband, however, often violence (dowry related or otherwise) including sexual violence, is inflicted other members of the family – how do the authors account for that? At the very least some acknowledgement of the fact is required.

Response: This is acknowledged in the limitations under discussion section.

5. Similarly, they continue to say, “Similarly, violence had a significant positive association with the educational level of girls and wealth index.” – however it is a negative relationship between violence and educational level – i.e. the higher the education, the lower the violence and the same with wealth. The way ‘positive association’ reads now it implies, the greater the wealth the more the violence inflicted – which is the opposite of what the authors go on to explain is happening.

Response: Authors apologize for their carelessness. The statement is modified highlighting a negative association in the revised manuscript.

6. The authors report, “Interestingly, the likelihood of any violence was 41 percent and 2.06 times more likely among adolescent girls whose marital duration was 2-3 years [OR: 1.41; CI: 1.19-1.66] and 4 years or more [OR: 2.06; CI: 1.60- 2.64] respectively, compared to those whose marital duration was less than or equal to one year.” Why is this interesting? Is it not obvious – the more exposure, the more likelihood of experiencing violence? What would be interesting would be to explore if the trajectory of violence is upward with time and yet the girls choose to remain in the marriage? Is there a tipping point at which the girl might take some action or complain? Probably the data does not allow the authors to answer these questions, but they are nevertheless more interesting.

Response: The word “Interestingly” is removed now. And yes, we agree with your comment that although the information is not available in UDAYA survey data, it would be interesting to explore if the trajectory of violence is upward with time and yet the girls choose to remain in the marriage.

7. In the discussion section, the authors report: “In the process of empowering women, the previous study indicates that parents appear to be replacing property inheritance to their daughters by giving them alternative transfers in the form of higher dowries (42). This might have resulted in husbands and family in-laws of empowered women not perpetrating any violence.” – that is a big generalisation – often dowries are taken over by the in-laws and the woman has little or no control over her wealth. What is the evidence that higher dowries succeed in empowering women – especially if the amount bestowed is lower than that demanded by the in-laws?

Response: Thank you for the comment. We agree that the statements can be misleading. The reported statements are removed now.

8. Furthermore, the authors report, “The finding is similar to a study based on the National Family Health Survey which noted that because of being forthright against male dominance and better reporting of incidences of domestic violence, Indian working women have a higher likelihood of being victims of violence (47).”

It is not clear what the authors are claiming from their finding that those in paid employment are more likely to be victims of domestic violence – is it because they are more willing to acknowledge and report violence or because they pose a threat to their partners on traditional gender roles and hence are vulnerable to partner violence?

Response: The statements are modified for better understanding as per suggestion. It reads now as “better reporting of incidences of domestic violence by which they pose a threat to their partners on traditional gender roles and hence, Indian working women are vulnerable to partner violence”.

Further, does this mean that having paid employment makes some women more vulnerable and not having enough economic empowerment makes other women more vulnerable? If in both situations some women are vulnerable – then there must be other factors that will determine who is more likely to be vulnerable with economic empowerment and who isn’t. On the one hand the authors seem to suggest in the Discussion that when women are more likely to report violence (either because they are economically empowered or have been married for a longer period) they are more likely to suffer violence, but do not mention why this might be the case. Are the findings suggesting that women who are empowered to speak out against violence are more likely to suffer future violence as a result – if so – this sends out a very negative message as it does not support more empowerment for women – perhaps the authors might need to carefully caveat this finding.

Response: We agree with the reviewer’s concerns. The following statements have been added to clarify this. “The current finding may be explained by the fact that economically empowered women who have more gender-conservative partners may have increased risk of experiencing violence especially, when they become less willing to conform to patriarchal norms in the household.” Relevant citation is provided.

Further they go on to state that higher education status and social class act as protective factors and such women are less likely to be victims of violence. I am not sure the authors have reconciled these contrary findings effectively.

Response: The discussion is revised accordingly. The added explanation reads as “The contrary findings in this study in terms of female labour-force participation increasing the risk and socioeconomic status reducing the risk of IPV highlight the importance of resource distribution and power imbalances within the family that may lead to or protect adolescent girls from IPV.” A related reference has been added too.

9. A few grammatical issues need addressing. For example,

a. There are no page numbers for giving exact location, but in the results section the authors report “For instance, married girls faced more violence to increase the length of the marriage.” – which does not sound correct – Suggest rephrase.

Response: The sentence is rephrased and corrected now.

b. “A dearth of literature repeatedly found educational attainment and higher social status as major factors to reduce the likelihood of experiencing domestic violence in developed as well as developing countries (56–59)”. How can a dearth of literature (which means lack of literature) repeatedly find ….something?

Response: The sentence modified by rephrasing the word “dearth” with “wealth”

c. The authors state, “This study is based on sample of married girls from both the states which is 5,226 which is representative of 15-19 married girls at state level.” – perhaps they mean sample representing married girls between the age group of 15 to 19 years.

Response: The statement is modified now for better understanding.

10. One big gap in the paper is the relationship between violence and whether the woman has been able to produce children, preferably male offspring which might be related to or completely separate to demands for dowry. Were the women asked whether they felt their victimisation by their partner was due to unfulfilled demands for dowry and/or other factors?

Response: This information was not collected in UDAYA.

11. Did the dataset not have any information about dowry related domestic violence not perpetrated by the husband/partner? How do the authors propose to deal with this lacuna?

Response: This information was not available.

I do believe that the paper can make an important contribution if the authors work on developing a more nuanced conceptual framework and hypothesise which factors provide effective protection or which combination of factors make women more vulnerable to victimisation. A defined theoretical framework to make better sense of the data would make a significant contribution to the current literature on the topic. This would also help the authors structure their answer the ‘so what’ question better in terms of concrete policy implications which in the current version are a bit imprecise.

Response: A more nuanced conceptual framework has been provided in the revised manuscript. The framework provides risk and protective factors of IPV. Discussion section has been revised substantially as per reviewer’s suggestions to explain the contrary findings and better understand the contexts.

---

## [Decision Letter · Decision Letter 2]

11 Aug 2022

PONE-D-21-13207R2Dowry demand and associated partner violence among married adolescent girls: A cross-sectional analytical study in IndiaPLOS ONE

Dear Dr. T.,

Thank you for submitting your manuscript to PLOS ONE. After careful consideration, we feel that it has merit but does not fully meet PLOS ONE’s publication criteria as it currently stands. Therefore, we invite you to submit a revised version of the manuscript that addresses the points raised during the review process. Please note that PLOS ONE does not copy edit accepted manuscripts (https://journals.plos.org/plosone/s/criteria-for-publication#loc-5). To that effect, please ensure that your revision is free of typos and grammatical errors. Please also ensure you have also stated whether you obtained consent from parents or guardians of the minors included in the study or whether the research ethics committee or IRB specifically waived the need for their consent.

We look forward to receiving your revised manuscript.

Kind regards,

Avanti Dey, PhD

Staff Editor

PLOS ONE

Journal Requirements:

Reviewers' comments:

Reviewer's Responses to Questions

**Comments to the Author**

1. If the authors have adequately addressed your comments raised in a previous round of review and you feel that this manuscript is now acceptable for publication, you may indicate that here to bypass the “Comments to the Author” section, enter your conflict of interest statement in the “Confidential to Editor” section, and submit your "Accept" recommendation.

Reviewer #3: All comments have been addressed

2. Is the manuscript technically sound, and do the data support the conclusions?

Reviewer #3: Yes

3. Has the statistical analysis been performed appropriately and rigorously? 

Reviewer #3: Yes

4. Have the authors made all data underlying the findings in their manuscript fully available?

Reviewer #3: Yes

5. Is the manuscript presented in an intelligible fashion and written in standard English?

Reviewer #3: Yes

6. Review Comments to the Author

Reviewer #3: (No Response)

7. PLOS authors have the option to publish the peer review history of their article (what does this mean?). If published, this will include your full peer review and any attached files.

Reviewer #3: No

---

## [Author Response · Author response to Decision Letter 2]

22 Aug 2022

Thank you for submitting your manuscript to PLOS ONE. After careful consideration, we feel that it has merit but does not fully meet PLOS ONE’s publication criteria as it currently stands. Therefore, we invite you to submit a revised version of the manuscript that addresses the points raised during the review process.

Response: Thank you for the opportunity to revise the manuscript. However, there were no further comments from the reviewer. The minor revision has been made as per suggestion.

Journal Requirements:

Response: The references are updated as per journal requirements.

Reviewers' comments:

Reviewer's Responses to Questions

Comments to the Author

1. If the authors have adequately addressed your comments raised in a previous round of review and you feel that this manuscript is now acceptable for publication, you may indicate that here to bypass the “Comments to the Author” section, enter your conflict of interest statement in the “Confidential to Editor” section, and submit your "Accept" recommendation.

Reviewer #3: All comments have been addressed

Response: Thank you for the recommendation.

---

## [Decision Letter · Decision Letter 3]

11 Oct 2022

PONE-D-21-13207R3Dowry demand and associated partner violence among married adolescent girls: A cross-sectional analytical study in IndiaPLOS ONE

Dear Dr. Muhammad,

Thank you for submitting your manuscript to PLOS ONE. After careful consideration, we feel that it has merit but does not fully meet PLOS ONE’s publication criteria as it currently stands. Therefore, we invite you to submit a revised version of the manuscript that addresses the points raised during the review process.

**Reviewer 3.**

I have no further comments except that the conceptual model needs to be updated and in line with the findings of the study. For example, non-Hindu category being only 15.9% of the sample and non significant in Tables 2 and 3, is presented as a risk factor in the model- Domestic violence is prevalent across religions and is not a characteristic of one particular religion

**Reviewer 5**

Thank you for the opportunity to review the manuscript entitled: “Dowry demand and associated partner violence among married adolescent girls: A cross-sectional analytical study in India”. The authors have tackled an important public health problem. Understanding how negative societal norms and cultures impact women’s risk of violence experience is important in the development of violence prevention programs and policies to mitigate these negative experiences, especially for young vulnerable women. However, I have some suggestions and recommendations that may help in strengthening the paper as outline below.

The authors have presented the conceptual framework for their study which seems rather simplistic. I expected more inter-linkages between the identifies clusters of risk factors or protective factors. As it is now, most factors only have direct paths to the outcome (IPV). What does literature say on link between Education status and attitudes that accept violence against women, decision making autonomy, employment etc for example. The other main concern is that their analytical approach does not seem to be consistent with their conceptual model.

Their “Data and Methods” section could be written in a more concise and clear way. The authors could briefly describe the original (UDAYA) study with references and then focus on describing data that they are using in this particular analysis. Otherwise, the whole first page of the “methods” section is confusing. For example, on the first page of the methods, they mention an effective sample size of 5226 and then on the second page the sample is 4893, with not so clear explanation on how the sample reduced to 4893. There is also a lot of repetitions of information in the section. The authors need to revise the section and make it more concise.

It is not necessary to include the mathematical definition of a logistic model. If authors really want to include it, then they should interpret all the terms in the model in the context of their data. The authors have just given a generic interpretation of the linear component of the logit model but do not explain the logit function of the model in the context of their binary outcomes. It is easier to just state in the data analysis section that logistic regression model will be used. They do not need to include the model.

Authors should make it explicit in Table 3 that they are presenting adjusted OR. Considering that their write up in the results section for the logistic regression only covers combined violence outcome and not the specific type of violence, I would suggest Table 3 focuses on that. They could then include the crude ORs, so a reader could look at both crude and adjusted ORs for main exposure variables.

The authors need to explain/justify in the methods section why the multivariable model included variables that were non-significant in the Chi-square test (Table 2).

Does the original UDAYA study have data on partner characteristics eg age of partner as these could also have impact on the power dynamics within the Household and impact on risk of IPV. If this data is available, authors need to adjust for these in the model. Another crucial variable that they could look at which would be more informative on cumulative risk of IPV (lifetime experience of IPV) than marriage duration is the age of participant at marriage. This could be derived from age of participant and marriage duration (assuming the participants are in their first marriage).

The last sentence on second page of the discussion section: “The results are in agreement with an earlier study in India which found that women married for five years and longer were more likely to be beaten by their husbands [53].”, is rather misleading considering what is being measured in the study is ‘lifetime experience of IPV” as such the authors are looking at cumulative experience of IPV. I would suggest this sentence is revised to: “The results are in agreement with an earlier study in India which found that women married for five years and longer were more likely to have been beaten by their husbands [53].”, to reflect cumulative risk due cumulative exposure.

 Please submit your revised manuscript by November 21, 2022. If you will need more time than this to complete your revisions, please reply to this message or contact the journal office at plosone@plos.org. Please include the following items when submitting your revised manuscript:A rebuttal letter that responds to each point raised by the academic editor and reviewer(s). You should upload this letter as a separate file labeled 'Response to Reviewers'.A marked-up copy of your manuscript that highlights changes made to the original version. You should upload this as a separate file labeled 'Revised Manuscript with Track Changes'.An unmarked version of your revised paper without tracked changes. You should upload this as a separate file labeled 'Manuscript'.

We look forward to receiving your revised manuscript.

Kind regards,

Yandisa Sikweyiya, PhD

Guest Editor

PLOS ONE

Reviewers' comments:

Reviewer's Responses to Questions

**Comments to the Author**

1. If the authors have adequately addressed your comments raised in a previous round of review and you feel that this manuscript is now acceptable for publication, you may indicate that here to bypass the “Comments to the Author” section, enter your conflict of interest statement in the “Confidential to Editor” section, and submit your "Accept" recommendation.

Reviewer #3: All comments have been addressed

Reviewer #5: (No Response)

2. Is the manuscript technically sound, and do the data support the conclusions?

Reviewer #3: Yes

Reviewer #5: Partly

3. Has the statistical analysis been performed appropriately and rigorously? 

Reviewer #3: Yes

Reviewer #5: Yes

4. Have the authors made all data underlying the findings in their manuscript fully available?

Reviewer #3: (No Response)

Reviewer #5: Yes

5. Is the manuscript presented in an intelligible fashion and written in standard English?

Reviewer #3: Yes

Reviewer #5: No

6. Review Comments to the Author

Reviewer #3: I have not further comments except that the conceptual model need to be updated and in line with the findings of the study. For example, non-Hindu category being only 15.9% of the sample and non significant in Tables 2 and 3, is presented as a risk factor in the model- Domestic violence is prevalent across religions and is not a characteristic of one particular religion.

Reviewer #5: Thank you for the opportunity to review the manuscript title “ Dowry demand and associated partner violence among married adolescent girls: A cross-sectional analytical study in India”. The authors have tackled an important public health problem. Understanding how negative societal norms and cultures impact women’s risk violence experience are important in the development of violence prevention programs and policies to mitigate these negative experiences, especially for young vulnerable women. However, I have some suggestions and recommendations that may help in strengthening the paper as outline below.

The authors have presented the conceptual framework for their study which seems rather simplistic. I expected more inter-linkages between the identifies clusters of risk factors or protective factors. As it is now, most factors only have direct paths to the outcome (IPV). What does literature say on link between Education status and attitudes that accept violence against women, decision making autonomy, employment etc for example. The other main concern is that their analytical approach does not seem to be consistent with their conceptual model.

Their “Data and Methods” section could be written in a more concise and clear way. The authors could briefly describe the original (UDAYA) study with references and then focus on describing data that they are using in this particular analysis. Otherwise the whole first page of the “methods” section is confusing. For example, on the first page of the methods, they mention an effective sample size of 5226 and then on the second page the sample is 4893, with not so clear explanation on how the sample reduced to 4893. There is also a lot of repetitions of information in the section. The authors need to revise the section and make it more concise.

It is not necessary to include the mathematical definition of a logistic model. If authors really want to include it, then they should interpret all the terms in the model in the context of their data. The authors have just given a generic interpretation of the linear component of the logit model but do not explain the logit function of the model in the context of their binary outcomes. It is easier to just state in the data analysis section that logistic regression model will be used. They do not need to include the model.

Authors should make it explicit in Table 3 that they are presenting adjusted OR. Considering that their write up in the results section for the logistic regression only covers combined violence outcome and not the specific type of violence, I would suggest Table 3 focuses on that. They could then include the crude ORs, so a reader could look at both crude and adjusted ORs for main exposure variables.

The authors need to explain/justify in the methods section why the multivariable model included variables that were non-significant in the Chi-square test (Table 2).

Does the original UDAYA study have data on partner characteristics eg age of partner as these could also have impact on the power dynamics within the Household and impact on risk of IPV. If this data is available, authors need to adjust for these in the model. Another crucial that they could look at which would be more informative on cumulative risk of IPV (life time experience of IPV) than marriage duration is the age of participant at marriage. This could be derived from age of participant and marriage duration (assuming the participants are in their first marriage).

The last sentence on second page of the discussion section : “The results are in agreement with an earlier study in India which found that women married for five years and longer were more likely to be beaten by their husbands [53].” , is rather misleading considering what is being measured in the study is ‘life time experience of IPV” as such the authors are looking at cumulative experience of IPV. I would suggest this sentence is revised to : “The results are in agreement with an earlier study in India which found that women married for five years and longer were more likely to have been beaten by their husbands [53].”, to reflect cumulative risk due cumulative exposure.

7. PLOS authors have the option to publish the peer review history of their article (what does this mean?). If published, this will include your full peer review and any attached files.

Reviewer #3: No

Reviewer #5: No

---

## [Author Response · Author response to Decision Letter 3]

23 Nov 2022

Response to reviewer comments

Reviewer 3

I have no further comments except that the conceptual model needs to be updated and in line with the findings of the study. For example, non-Hindu category being only 15.9% of the sample and non-significant in Tables 2 and 3, is presented as a risk factor in the model- Domestic violence is prevalent across religions and is not a characteristic of one particular religion

Response: Thank you for the additional comment and the recommendation. The conceptual framework has been modified as per suggestion.

Reviewer 5

Thank you for the opportunity to review the manuscript entitled: “Dowry demand and associated partner violence among married adolescent girls: A cross-sectional analytical study in India”. The authors have tackled an important public health problem. Understanding how negative societal norms and cultures impact women’s risk of violence experience is important in the development of violence prevention programs and policies to mitigate these negative experiences, especially for young vulnerable women. However, I have some suggestions and recommendations that may help in strengthening the paper as outline below.

The authors have presented the conceptual framework for their study which seems rather simplistic. I expected more inter-linkages between the identifies clusters of risk factors or protective factors. As it is now, most factors only have direct paths to the outcome (IPV). What does literature say on link between Education status and attitudes that accept violence against women, decision making autonomy, employment etc for example. The other main concern is that their analytical approach does not seem to be consistent with their conceptual model.

Response: Many thanks for the insightful comment. The conceptual framework is modified after incorporating the suggestions. The low level of education is kept as a risk factor of IPV and the higher level of education is included as a protective factor along with paid work for attitudes of accepting violence and decision making autonomy.

Their “Data and Methods” section could be written in a more concise and clear way. The authors could briefly describe the original (UDAYA) study with references and then focus on describing data that they are using in this particular analysis. Otherwise, the whole first page of the “methods” section is confusing. For example, on the first page of the methods, they mention an effective sample size of 5226 and then on the second page the sample is 4893, with not so clear explanation on how the sample reduced to 4893. There is also a lot of repetitions of information in the section. The authors need to revise the section and make it more concise.

Response: Thanks for the comment. Authors have made changes in the data and methods section and removed the repetitions of the information.

It is not necessary to include the mathematical definition of a logistic model. If authors really want to include it, then they should interpret all the terms in the model in the context of their data. The authors have just given a generic interpretation of the linear component of the logit model but do not explain the logit function of the model in the context of their binary outcomes. It is easier to just state in the data analysis section that logistic regression model will be used. They do not need to include the model.

Response: Thanks for the comment. Authors have removed mathematical definition of the logistic model.

Authors should make it explicit in Table 3 that they are presenting adjusted OR. Considering that their write up in the results section for the logistic regression only covers combined violence outcome and not the specific type of violence, I would suggest Table 3 focuses on that. They could then include the crude ORs, so a reader could look at both crude and adjusted ORs for main exposure variables.

Response: Dear reviewer, thank you for this comment. The authors have now mentioned that in the result section only combined violence was interpreted and not eh specific type of violence. Moreover, the ORs presented are adjusted odds ratios; however, the “adjusted” term was not earlier used. The authors have used the “aOR” for the adjusted odds ratio. 

The authors need to explain/justify in the methods section why the multivariable model included variables that were non-significant in the Chi-square test (Table 2).

Response: Even though background characteristics(basic) were independently in-significant in the current study, the author still wants to consider age, relation & wealth index variables because adjusting these background characteristics may affect the significance of other determining/independent variables.

Does the original UDAYA study have data on partner characteristics eg age of partner as these could also have impact on the power dynamics within the Household and impact on risk of IPV. If this data is available, authors need to adjust for these in the model. Another crucial variable that they could look at which would be more informative on cumulative risk of IPV (lifetime experience of IPV) than marriage duration is the age of participant at marriage. This could be derived from age of participant and marriage duration (assuming the participants are in their first marriage).

Response: In the study, respondent age is from 15-19 Years which is already a very small span. Age at marriage is not normally distributed and there was high multi-collinearity (high VIF) between marriage duration and age at marriage. Hence, the authors did not consider including the age at marriage. The bivariate analysis showed no statistical significance in the association of partner age and IPV, thus not considered as well.

The last sentence on second page of the discussion section: “The results are in agreement with an earlier study in India which found that women married for five years and longer were more likely to be beaten by their husbands [53].”, is rather misleading considering what is being measured in the study is ‘lifetime experience of IPV” as such the authors are looking at cumulative experience of IPV. I would suggest this sentence is revised to: “The results are in agreement with an earlier study in India which found that women married for five years and longer were more likely to have been beaten by their husbands [53].”, to reflect cumulative risk due cumulative exposure.

Response: The comment is incorporated now. The statement is modified as per suggestion.

---

## [Decision Letter · Decision Letter 4]

6 Jun 2023

PONE-D-21-13207R4Dowry demand and associated partner violence among married adolescent girls: A cross-sectional analytical study in IndiaPLOS ONE

Dear Dr. Muhammad,

Thank you for submitting your manuscript to PLOS ONE. After careful consideration, we feel that it has merit but does not fully meet PLOS ONE’s publication criteria as it currently stands. Therefore, we invite you to submit a revised version of the manuscript that addresses the points raised during the review process.

We look forward to receiving your revised manuscript.

Kind regards,

Obasanjo Bolarinwa

Academic Editor

PLOS ONE

Reviewers' comments:

Reviewer's Responses to Questions

**Comments to the Author**

1. If the authors have adequately addressed your comments raised in a previous round of review and you feel that this manuscript is now acceptable for publication, you may indicate that here to bypass the “Comments to the Author” section, enter your conflict of interest statement in the “Confidential to Editor” section, and submit your "Accept" recommendation.

Reviewer #6: (No Response)

Reviewer #7: (No Response)

2. Is the manuscript technically sound, and do the data support the conclusions?

Reviewer #6: Yes

Reviewer #7: Partly

3. Has the statistical analysis been performed appropriately and rigorously? 

Reviewer #6: Yes

Reviewer #7: Yes

4. Have the authors made all data underlying the findings in their manuscript fully available?

Reviewer #6: Yes

Reviewer #7: Yes

5. Is the manuscript presented in an intelligible fashion and written in standard English?

Reviewer #6: Yes

Reviewer #7: Yes

6. Review Comments to the Author

Reviewer #6: Although suggested by reviewer 3 to present the odds ratios for crude ORs, the authors have not modified accordingly, additionally, I would recommend presenting the regression results by various forms of IPV and dowry demand (maybe in a separate table or in appendix)

In response to comments received from reviewer 3, the authors say that they included the background characteristics despite the variables being insignificant as they might have impact on IPV after being adjusted, however, in the next response the authors refute the idea of including age of partner claiming those variables to be insignificantly associated. I feel the authors should consider including partner’s age as it is an important determinant of IPV.

Abstract:

Line “The present study aimed to examine the association of dowry …” should be corrected to association between ‘. The conclusion section is unclear in terms of language and clarity.

The authors are advised to look thoroughly for grammatical errors.

Introduction- I do not understand what the authors mean by the statement “this study may contribute to how gender-specific risk and protective factors…” . I don’t feel that the analysis and context of the study in any-way related to both gender or gender-specific risk or protective factor.

Manuscript

In the introduction section the authors hypothesize that demand for dowry is associated with increased violence among married adolescent girls. However, the authors make no, make no mention of the rejection or acceptance of hypothesis. The authors should clearly mention it in the manuscript.

In data and methods, the authors mention that dowry demand question was asked as “Dowry demand by in-laws” however, in abstract and in results section the authors write as “dowry was demanded by their husbands”. The authors should check for such errors and correct it throughout the text.

The data section should be more concise.

As per the title of the manuscript, I expected a focus on the association and impact of dowry on the IPV which I could not find. An extensive elaborate discussion of other predictors impacting IPV diverts from the central idea of the paper. I would recommend the authors consider revising the discussion and an emphasis is given on the dowry which is the core idea of this paper. Additionally, the discussion should include findings from this study which now seems just to be a theoretical evidence from other literature.

Additionally, I could not find anything related to dowry and its association with dowry in the conclusion section which should have been as per the focus of the study.

I do not understand what the authors mean when they say” A large proportion reporting dowry may be explained by the fact that many of them do not consider dowry as a repressive practice but a rightful share…” If the respondent views the dowry as their rightful share, the proportion reporting dowry should be low. If so, what I understand and what is mentioned are two different sides of the coin.

Reviewer #7: Responses to 2, 3, 5 reflect that overall the manuscript is well written and analyses are strong, but improvements are needed on all of these areas.

7. PLOS authors have the option to publish the peer review history of their article (what does this mean?). If published, this will include your full peer review and any attached files.

Reviewer #6: No

Reviewer #7: No

---

## [Author Response · Author response to Decision Letter 4]

25 Feb 2024

Response to comments

Reviewer #6

Thank you very much for the opportunity to review this manuscript on IPV among married adolescent girls in India. This manuscript makes important contributions to understanding married adolescent girls’ risk of IPV related to dowry demand, acceptance of IPV, and household decision-making power. The large sample of married adolescents in India and the findings on the relationship between the factors (dowry demand, acceptance of IPV, and household decision-making power) and IPV are the strongest contributions of this paper to the literature on IPV risk. As such, major revisions focusing on these factors throughout the paper and analyses would greatly strengthen this manuscript and its value to the field.

Abstract: 

Last sentence of “Background” missing word “using [a] large dataset”.

Response: Dear reviewer, thank you for your feedback. We have added the missing word "using a large dataset" in the revised manuscript.

Last sentence of “Conclusion” – suggest revising wording to clarify what “say” means. Is it decision-making power or agency? 

Response: Dear reviewer, thank you for your comment. The authors have addressed the concern by revising the conclusions part of the abstract. 

Background: 

The background provided on dowry demand and hypothesized relationship with IPV is strong and some of the context of gender inequity is provided that helps frame why they authors included the other exposure variables. However, the background on the other exposure variables and the focus on these in the analysis needs to be reflected in this background more. As commented below about Figure 1, the background section could be strengthened with a more structured approach to discussing the main exposure variables that will be focused on in this paper and are contributions to the literature: dowry demand (justification is already quite clear), perception of wife-beating, decision-making on work and household purchases. 

The first paragraph talks about IPV and dowry broadly/globally, but then provides very specific information about legal codes in India. The setting of your paper needs to be introduced more formally – whether you want to frame it around South Asia or India is your choice. I would recommend introducing the setting in this first paragraph – talk about IPV globally and then particularly in India. Next paragraph, talk about dowry broadly and then in India specifically. 

Is there a national estimate for how common dowries are? Please provide, if so, as this would provide a parallel description with the IPV estimates.

Last sentence in paragraph 3: I recommend rewording for clarity, “…could be a major determinant for domestic violence victimization.”

Fifth paragraph: Please define dowry deaths. Also, please reword this sentence for clarity, “However, little is known about the nature of the problem and married adolescent girls' experiences of dowry demand during and early years of their marriage resulting in some form of violence”. Particularly what is meant by “during and early years of their marriage” and what is the following clause attached to in this sentence: “some form of violence.”

Please correct small grammatical errors throughout. 

Response: We apologize for these errors and have corrected them in the updated version of the manuscript. We now provide a national estimate of dowry demand in India in the revised introduction part. We have implemented your other suggestions, reorganizing the manuscript to give prominence to global IPV studies, followed by a focus on South Asia. The paper now centers on two key contributions: examining perceptions of wife-beating and exploring decision-making dynamics related to work and household purchases. The introduction introduces the setting more formally, first discussing IPV globally and then specifically in India. The discussion on dowry follows a recommended structure in the updated version of the manuscript.

Figure 1: Overall, this conceptual framework does not add a lot of value to this manuscript and may cause more confusion than benefit. Because the analysis does not align with this framework, the authors may benefit more from very clearly justifying in the background the inclusion of the key hypothesized exposures for IPV: dowry demand (justification is already quite clear), perception of wife-beating, decision-making on work and household purchases. 

If the authors do decide to keep it, here are some specific concerns: 1) the analysis does not reflect the relationship presented here. 2) Paid work and higher education should appear further to the left as it proceeds justification for wife beating and decision-making autonomy. 3) What is meant by box that says, “Can be catalized by…”? Meaning dowry demand by in-laws plus being of lower wealth, lower caste, and rural may make IPV more likely? If you’re describing a causal ordered process, seems like there should be an arrow lining dowry demand to your other box on rural, caste, and income, and then continue to IPV. Alternatively, is caste/income/rural a driver of dowry demand? If so, that box should come before the dowry demand one on that line. 4) Same comment for “Justified perception over wife beating” (this should also be corrected to read: “Perceived justification for wife beating”). 

Response: We appreciate this suggestion and have now removed the conceptual framework to avoid confusion among readers!

Methods

Data

Paragraph 1 needs further refinement for clarity. Seems to me that the necessary information is here the order of it just needs to be adjusted. My recommendation would be to revise the first paragraph after the third sentence so that information is provided in the following order (mainly rearrange sentences and take out anything extra): 

1. Sampling frame

2. Total sample of PSUs

3. Factors included in stratified sampling and their hierarchy 

4. What was done for rural villages

5. What was done for urban villages

Then I would recommend making the information about the five categories of respondents a separate paragraph. 

Response: Thank you for the comment! Per comments from Reviewer #7, we have reduced the content of the data section to make it concise and provided a citation for further information on survey sampling procedures.

Measures

Outcome variable - First two sentences are repetitive. Delete first sentence and edit the second sentence to read, “Emotional violence was measured using the question “Did your husband ever do something to humiliate you in front of others or threatened you to hurt or harm someone close to you?” and the response was coded as no (0) and yes (1). What were the original response options on the violence outcome questions before recoding?

Response: Thank you for your thoughtful suggestions. And the violence variable consists of 7 questions. If the respondent ever expressed any of the following and responded "sometimes" or "often" in the last 12 months, we recorded ever experiencing those into 1 as 'yes' and never experiencing them into 0 as ‘no.’

During the survey, the following questions were asked:

1. Did your husband ever do any of the following to you: YES/NO

2. How often did this happen in the last 12 months: never, sometimes, or often? a) slap you? b) twist your arm or pull your hair? c) push you, shake you, or throw something at you? d) punch you with his fist or with something that could hurt you? e) kick you, drag you, or beat you up? f) try to choke you or burn you on purpose? g) threaten or attack you with a knife, gun, or any other weapon?: NEVER/SOMETIMES/OFTEN

Exposure variable – This paragraph could use some tightening (make more succinct) I would recommend starting the paragraph with the question(s) that were asked about dowry demand. Were these two separate questions or combined as written in the text? Also, what were the response options? Then state how the variable was created. 

Response: Dear Reviewer, the author has implemented your suggestions. Dowry demand by in-laws was assessed by asking, “Has anybody in your husband’s family ever said that the dowry/gift/cash you brought was too little or that you did not bring anything?” the response was recoded as 1 ('yes') and 0 ('no') otherwise.

The question was singular, with only one inquiry and response options limited to 'Yes' and 'No,' as demonstrated above. This is clarified now in the updated version.

I would recommend that sentences 3-5 of the “exposure variables” paragraph be removed, as these should be already established in your introduction. I would recommend removing and go right into defining your other exposure variables. The definition of each variable would be clarified if a paragraph for each variable was included. That paragraph could briefly include the question that was asked, where did that measure come from with citation (previously validated? Used elsewhere? New and if so, how it was developed?), the response options, and how the variable was created/coded. 

Response: Dear Reviewer, the author has implemented your suggestions. We appreciate your suggestions.

For the variable about decisions regarding household purchases, it is important that throughout the paper this be referred to as household “purchases” (not “purchase”) because the current phrasing throughout the paper makes this sounds like it is about the decision to buy a house or not, rather than everyday household purchases. This edit is needed throughout the paper and tables.

Response: Thank you for pointing out. Made the changes. 

The inquiry pertained to household purchases: "Who primarily decides about major household purchases? Do you alone decide, jointly decide with others, or do others alone decide?" In the Results and other sections, it was corrected from "purchase" to accurately reflect the question.

Statistical analysis

Is there a citation for the UDAYA dataset documentation/information about weights? 

Response: Yes, the relevant citation for the UDAYA dataset documentation and the survey weights is provided now in the data section.

I recommend that the statement about terming late adolescents to be removed because this term is not used at all in the majority of the paper. 

Response: Dear reviewer, thank you for the comment. The authors have addressed the concern by removing the term "late adolescents" from the manuscript.

What were the adjusted models controlling for? Or were they only adjusted for clustering and stratum effects? Rather than considering marital duration as an exposure variable, I would be more inclined to control for this variable. It seems logical and well-established that longer duration of marriage would provide more opportunity and increased risk for violence within marriage so feels less like a novel finding and more like an important variable to control for. Similarly, age (exclude if collinear with marriage duration), urban/rural residence, religion, caste, education, wealth, and state are all demographic characteristics with known patterns of risk for IPV that are better suited as covariates. If the authors were to control for these variables, it would help to isolate the estimated effects of dowry demand, perception of wife-beating, decision-making on work and household purchases on IPV risk, which is the strongest contribution of this paper. 

Response: Thank you for your suggestions. To clarify, our models were adjusted for these demographic characteristics to isolate the estimated effects of dowry demand, perception of wife-beating, and decision-making on work and household purchases on IPV risk. These are clarified as table notes in the revised version. The tables are modified per reviewer suggestions in the updated version.

Did you consider controlling for age at marriage? There is evidence to suggest that risk for violence varies (as well as many of the exposure variables) by how young the adolescent is when she marries. Along these lines, did you consider (or was it possible) to control for the age difference between the wife and husband? Again, evidence suggests that this might be an important indicator of the power imbalance between husband and wife, and therefore be related to increased risk for violence. 

Response: Dear reviewer, in light of your suggestion and considering the recommendations of other reviewers, we revisited the analysis and included controls for the age of the spouse. Thank you for bringing this to our attention, and we value your input in enhancing the robustness of our study.

Results

The contribution of this paper are the results related to the estimated effect of dowry demand, perception of wife-beating, decision-making on work and household purchases on IPV risk. As such, I recommend keeping Tables 1 and 2 as they are (as well as the presentation of results), and modifying Table 3 to include analyses that are focused on these 4 exposures with aOR results that are adjusted for the sociodemographic factors. 

Response: Dear reviewer, thanks for the comment. The authors have amended the table-3 (and table 4 in the updated version) and the results as per your valuable suggestion. 

More detailed comments on current manuscript results section:

The sentence is a bit misleading or at the very least, confusing: “Moreover there was a positive relationship between type of violence and duration…” Put more simply, the authors could say what types of violence are associated with marital duration and exclude the example. 

Response: As per your previous comment we have shortened the results for table 3 (and table 4 in the updated version), therefore, this sentence is not present in the manuscript now. However, thank you so much for your in-depth suggestion. 

Paragraph 3, sentence 5: Recommend revising for clarity. In particular, I would recommend rephrasing to “adolescent girls whose decisions about their work are made by others” (similar edit to other instances in this paragraph referring to “decisions taken”) and removing the work “only” to remove the interpretation in this statement. 

Response: Dear reviewer, thanks for the comment. The authors have amended the sentences as per your comment. 

Par 3, sentence 7: It is a bit odd to interpret the 1.41 aOR as a percent and the 2.06 as a likelihood in the same sentence. I would recommend choosing one approach for both within a sentence. 

Response: As per your previous comment we have shortened the results for table 3 (and table 4 in the updated version), therefore, this sentence is not present in the manuscript now. However, thank you so much for your in-depth suggestion, we have cautioned this while interpreting the results in the updated manuscript. 

Results related to caste: clarify that “other castes” does not include OBC. 

Response: As per your previous comment we have shortened the results for table 3, therefore, this sentence is not present in the manuscript now. 

Table 3: footnotes should clarify what the aOR models adjusted for. 

Response: Dear reviewer, thanks for the comment. The authors have amended the footnote of the table 3 (and table 4 in the updated version) as per your suggestion. 

Discussion

Paragraph 2: It seems likely that women who have experienced IPV are more likely to perceive violence is justified because of the ways that IPV perpetrators rationalize and tell their victims that they are deserving of that violence, rather than their pre-existing acceptance of inequitable gender roles being the cause of their husbands’ violence. Many survivors of IPV are told so many times by their perpetrators that they are deserving of the abuse that the survivor begins to believe it themselves, or at least accepts this perception to cope with the need to remain married to the person. The discussion of this factor might benefit from a more nuanced discussion of the direction of this relationship being interrogated and bring in literature to help frame this discussion. This discussion could replace some of the discussion about the sociodemographic variables being associated with IPV that are less of a strong contribution to the literature. 

Response: After the second paragraph in the discussion section, we have touched upon the strategies employed by IPV perpetrators to convince survivors that they are deserving of the abuse they endure. We further emphasize the p

---

## [Decision Letter · Decision Letter 5]

27 Aug 2024

PONE-D-21-13207R5Dowry demand, perception of wife-beating, decision making power and associated partner violence among married adolescent girls: A cross-sectional analytical study in IndiaPLOS ONE

Dear Dr. Muhammad,

Thank you for submitting your manuscript to PLOS ONE. After careful consideration, we feel that it has merit but does not fully meet PLOS ONE’s publication criteria as it currently stands. Therefore, we invite you to submit a revised version of the manuscript that addresses the points raised during the review process.

We look forward to receiving your revised manuscript.

Kind regards,

Obasanjo Afolabi Bolarinwa

Academic Editor

PLOS ONE

Journal Requirements:

Additional Editor Comments:

Please kindly review the manuscript for the last time to be scientifically judged.

Reviewers' comments:

Reviewer's Responses to Questions

**Comments to the Author**

1. If the authors have adequately addressed your comments raised in a previous round of review and you feel that this manuscript is now acceptable for publication, you may indicate that here to bypass the “Comments to the Author” section, enter your conflict of interest statement in the “Confidential to Editor” section, and submit your "Accept" recommendation.

Reviewer #8: (No Response)

2. Is the manuscript technically sound, and do the data support the conclusions?

Reviewer #8: Yes

3. Has the statistical analysis been performed appropriately and rigorously? 

Reviewer #8: Yes

4. Have the authors made all data underlying the findings in their manuscript fully available?

Reviewer #8: Yes

5. Is the manuscript presented in an intelligible fashion and written in standard English?

Reviewer #8: Yes

6. Review Comments to the Author

Reviewer #8: Background

Under the first sentence in the 2nd paragraph under "Background", the sentence "In South Asia, IPV is a complex challenge that is related to cultural dynamics." could be strengthened further by stating some of these cultural dynamics and possibly stating references that validates these cultural determinants with IPV.

In the subsequent sentence in the same paragraph, it would be great to start the sentence with an opening that establishes the role of personal experiences and societal factors in amplifying the likelihood of IPV. This can come just before the sentence "Research links IPV in the region, including India....".

Furthermore, within the same paragraph (sentence before last), the use of the word "ecology of IPV" seems vague - consider alternative terms like "prevalence".

Last sentence in the second paragraph, "Additionally, interventions, such as couple-based programs, have been explored for the primary prevention of IPV in India, emphasizing the need for effective strategies to address this issue" provided valuable perspective. However, the author should either expand on these interventions briefly here or move this entire sentence to the Conclusion session and share some existing interventions as suggested here.

Data & Method

I would re-label this section to "Methodology" and then the "data' to 'Overall Approach' - First paragraph and then "Sampling Strategy" - starting with the 2nd paragraph. Under the first paragraph, consider including a sentence briefly describing what the UDAYA project is about.

Variable description -

Include a summary paragraph that briefly described the types of variables analysed (in this case outcome & exposure variable) and a short rationale for focusing on each.

Statistical Analysis -

In line 3, you established that association between the binary 'outcome variable' and 'explanatory variables' were conducted. However, in your description of variables, you used 'exploratory variable' - there is need for consistency.

Discussion -

Last sentence in paragraph 5 should be reworded for clarity from "The present study in concordance with this found that

higher education and upper social class in the Indian caste hierarchy of married adolescent girls were protective factors against partner violence" to "The findings from this study which establishes that higher education and upper social class in the Indian caste hierarchy of married adolescent girls were protective factors against partner violence, is consistent with existing literature.

The subsequent paragraph (2nd sentence seem incomplete: "Owing to the cross-sectional nature of this study, no causal relationships could be established"...add "between X variables and Y variables". You may decide to stratify with of the causal factors were worse off between the personal, social, cultural or otherwise.

7. PLOS authors have the option to publish the peer review history of their article (what does this mean?). If published, this will include your full peer review and any attached files.

Reviewer #8: No

---

## [Author Response · Author response to Decision Letter 5]

17 Sep 2024

Response to comments

Reviewer #8

Background

Under the first sentence in the 2nd paragraph under "Background", the sentence "In South Asia, IPV is a complex challenge that is related to cultural dynamics." could be strengthened further by stating some of these cultural dynamics and possibly stating references that validates these cultural determinants with IPV.

Response: We appreciate your comment! We have added more details and validating references per suggestion. 

In the subsequent sentence in the same paragraph, it would be great to start the sentence with an opening that establishes the role of personal experiences and societal factors in amplifying the likelihood of IPV. This can come just before the sentence "Research links IPV in the region, including India....".

Response: We have updated the text and elaborated on the role of personal experiences and societal factors in amplifying the likelihood of IPV. 

Furthermore, within the same paragraph (sentence before last), the use of the word "ecology of IPV" seems vague - consider alternative terms like "prevalence".

Response: We have replaced the word with “prevalence”.

Last sentence in the second paragraph, "Additionally, interventions, such as couple-based programs, have been explored for the primary prevention of IPV in India, emphasizing the need for effective strategies to address this issue" provided valuable perspective. However, the author should either expand on these interventions briefly here or move this entire sentence to the Conclusion session and share some existing interventions as suggested here.

Response: We have expanded on these interventions in the updated manuscript.

Data & Method

I would re-label this section to "Methodology" and then the "data' to 'Overall Approach' - First paragraph and then "Sampling Strategy" - starting with the 2nd paragraph. Under the first paragraph, consider including a sentence briefly describing what the UDAYA project is about.

Response: Thank you for your comment! We have updated the section heading and added a brief description of UDAYA project under the first paragraph on the Overall Approach.

Variable description -

Include a summary paragraph that briefly described the types of variables analysed (in this case outcome & exposure variable) and a short rationale for focusing on each.

Response: We have added a description of types of variables analyzed and a short rationale for each.

Statistical Analysis -

In line 3, you established that association between the binary 'outcome variable' and 'explanatory variables' were conducted. However, in your description of variables, you used 'exploratory variable' - there is need for consistency.

Response: We have updated the text and used the word “explanatory” throughout the manuscript.

Discussion -

Last sentence in paragraph 5 should be reworded for clarity from "The present study in concordance with this found that higher education and upper social class in the Indian caste hierarchy of married adolescent girls were protective factors against partner violence" to "The findings from this study which establishes that higher education and upper social class in the Indian caste hierarchy of married adolescent girls were protective factors against partner violence, is consistent with existing literature.

Response: Incorporated this suggestion in the updated version. Thank you for pointing this out!

The subsequent paragraph (2nd sentence seem incomplete: "Owing to the cross-sectional nature of this study, no causal relationships could be established"...add "between X variables and Y variables". You may decide to stratify which of the causal factors were worse off between the personal, social, cultural or otherwise.

Response: Thank you for your valuable feedback. We have revised the manuscript to specify the variables involved in the analysis. Instead of the general statement that "no causal relationships could be established," we now clarify that "no causal relationships between dowry demand, perception of wife-beating, decision making power and IPV could be established." Regarding the suggestion to stratify the causal factors, while this study did not explicitly examine the stratification of personal, social, and cultural factors influencing intimate partner violence (IPV), we have included a discussion of key studies in the literature that explore the comparative impact of these factors. This provides context and supports the interpretation of our findings within the broader framework of existing research.

---

## [Editor Report · Decision Letter 6]

7 Oct 2024

Dowry demand, perception of wife-beating, decision making power and associated partner violence among married adolescent girls: A cross-sectional analytical study in India

PONE-D-21-13207R6

Dear Dr. Muhammad,

We’re pleased to inform you that your manuscript has been judged scientifically suitable for publication and will be formally accepted for publication once it meets all outstanding technical requirements.

Kind regards,

Obasanjo Afolabi Bolarinwa

Academic Editor

PLOS ONE
---

## [Editor Report · Acceptance letter]

16 Oct 2024

PONE-D-21-13207R6 

PLOS ONE

Dear Dr. Muhammad, 

I'm pleased to inform you that your manuscript has been deemed suitable for publication in PLOS ONE. Congratulations! Your manuscript is now being handed over to our production team.

Kind regards, 

on behalf of

Mr Obasanjo Afolabi Bolarinwa 

Academic Editor

PLOS ONE